# Body mass index and critical care outcomes in hospitalized COVID-19 patients—A national cohort study

Yolanda Bonilla[1,2‡]*, Daniel High[1,2‡], Jose Acosta Rullan[1,2☯], Jude Tabba[1,2☯],
Richard Shalmiyev[1,2☯], Tanner Noris[1,2☯], Andrea Folds[2☯], Ana Martinez[2☯], Daniel Heller[1,2☯],
Raiko Diaz[1,2☯], Siddarth Kathuria[1,2☯], Prerna Sharma[1,2☯], Mauricio Danckers[1,2‡]

1 Division of Pulmonary and Critical Care Medicine, HCA Florida Aventura Hospital, Aventura, Florida,
United States of America, 2 Department of Internal Medicine, HCA Florida Aventura Hospital, Aventura,
Florida, United States of America

☯ These authors contributed equally to this work.
‡ These authors also contributed equally to this work.
* yolandabonilla.md@gmail.com

Hospital, George Washington University,
UNITED STATES OF AMERICA

**Peer Review History:** PLOS recognizes the
benefits of transparency in the peer review
process; therefore, we enable the publication
of all of the content of peer review and
author responses alongside final, published
articles. The editorial history of this article is
available here: https://doi.org/10.1371/journal.
pone.0329779

## Abstract

### Background

The COVID-19 pandemic caused significant global mortality. Obesity is associated
with worse COVID-19 outcomes. This study examined the relationship between BMI,
clinical interventions, and outcomes in hospitalized COVID-19 patients using pre-
vaccine national data.

### Methods

We conducted a retrospective cohort study using de-identified electronic health
records from the HCA Healthcare database, comprising 149 hospitals across 18 U.S.
states. Adults (≥18 years) hospitalized with confirmed SARS-CoV-2 infection between
March 1 and December 31, 2022, were included. The primary outcome was a com-
posite of in-hospital mortality or discharge to hospice, analyzed by BMI category.
Secondary outcomes included inpatient mortality, need for mechanical ventilation
or tracheostomy, duration of mechanical ventilation, and ICU (Intensive Care Unit)
length of stay.

### Results

Out of 38,321 hospital encounters, 21,996 met the inclusion criteria. Unadjusted
analyses showed no significant differences in rates of all-cause mortality or hos-
pice discharge across BMI categories. However, obese patients had higher rates of
mechanical ventilation (7.8% vs. 4.6%, $p < 0.001$), tracheostomy placement (1.2%
vs. 0.6%, $p < 0.001$), longer duration of mechanical ventilation (mean 11.5 ± 15.1

**Data availability statement:** The raw data cannot be shared publicly for two primary reasons: (1) it contains sensitive information that could potentially identify study participants, and (2) the data is owned by HCA Healthcare, whose legal and institutional policies prohibit the distribution of any part of the dataset. For inquiries related to data access in accordance with institutional regulations, please contact Dr. Christopher Ochner, Research Director for the HCA East Florida Division, at Christopher. Ochner@hcahealthcare.com.

**Funding:** The author(s) received no specific funding for this work.

**Competing interests:** The authors have declared that no competing interests exist.

**Abbreviations:** ACE-2, Angiotensin-Converting Enzyme 2; ALI, Acute Lung Injury; All-cause mortality, Death from any cause during the study period; ANOVA, Analysis of Variance; ARDS, Acute Respiratory Distress Syndrome; β, Beta Coefficient (used in regression analysis); BMI, Body Mass Index; COVID-19, Coronavirus Disease; CI, Confidence Interval; ECMO, Extracorporeal Membrane Oxygenation; FRC, Functional Residual Capacity; HCA, Hospital Corporation of America; HIV, Human Immunodeficiency Virus; IRB, Institutional Review Board; IL-4, Interleukin-4, IMV, Invasive Mechanical Ventilation; ICU, Intensive Care Unit; kg/m², Kilograms per Square Meter; LOS, Length of Stay; MCP-1, Monocyte ChemoattractantProtein 1; OR, Odds Ratio; PAI-1, Plasminogen Activator Inhibitor 1; RAAS, Renin-Angiotensin-Aldosterone System; RR, Relative Risk; RRT, Renal Replacement Therapy; SARS-CoV-2, Severe Acute Respiratory Syndrome Coronavirus 2; SDB, Sleep-Disordered Breathing; SD, Standard Deviation; SPSS, Statistical Package for the Social Sciences; STATA, Data Analysis and Statistical Software; TNFα, Tumor Necrosis Factor Alpha; U.S., United States; USA, United States of America; DNR/DNI, Do Not Resuscitate/ Do Not Intubate.

vs. $6.8 \pm 8.7$ days, $p < 0.001$), and longer ICU stays ($8.3 \pm 10.4$ vs. $5.1 \pm 6.4$ days, $p < 0.001$) than normal BMI patients. In adjusted analyses controlling for age, sex, race, ethnicity, and comorbidities, obesity was independently associated with increased odds of all-cause mortality or hospice discharge (OR 1.29, 95% CI: 1.08–1.55, $p < 0.05$), inpatient mortality (OR 1.67, 95% CI: 1.34–2.08, $p < 0.001$), need for invasive mechanical ventilation (OR 1.55, 95% CI: 1.31–1.82, $p < 0.001$), and tracheostomy placement (OR 1.57, 95% CI: 1.03–2.41, $p < 0.05$). Obesity was a significant predictor of longer duration of mechanical ventilation ($\beta = 3.67$ days, 95% CI: 1.28–6.06, $p < 0.001$) and ICU stay ($\beta = 2.90$ days, 95% CI: 2.08–3.72, $p < 0.001$).

## Conclusion

Obesity was independently associated with increased risk of adverse clinical outcomes among hospitalized COVID-19 patients. These findings highlight the importance of BMI as a prognostic factor in acute COVID-19 management.

---

## Background

The COVID-19 infection, which emerged in late 2019 and expeditiously turned into a global pandemic [1], had disrupted healthcare systems and substantially increased morbidity and mortality, resulting in a reported excess mortality of tens of millions [2–5]. Early in the pandemic, a vast majority of COVID-19-related deaths were attributed to the progression of respiratory illnesses to acute respiratory distress syndrome (ARDS) with a heightened inflammatory cytokine release state caused by the wild-type SARS-CoV-2 strain [6,7]. As the pandemic progresses, the virus has evolved into less lethal strains, primarily causing mild symptoms [8,9].

During the pandemic, identifying high-risk groups was critical, with obesity emerging as a key comorbidity linked to increased risk of intubation, ICU admission, and mortality [10–13]. Various mechanisms likely contribute to this association; obesity promotes chronic inflammation through macrophage infiltration of adipose tissue, leading to immune cell reprogramming and heightened inflammatory responses to infections, increasing susceptibility to cytokine storm [14–18].

Observational studies, including those by Kompaniyets et al. have shown that obesity is strongly linked to adverse outcomes, with higher BMI associated with increased risk of ICU admission, mechanical ventilation, and death [11,19–27]. A smaller retrospective cohort study by Anderson et al. similarly found a dose-dependent relationship between BMI class and a composite endpoint of death or intubation, with the effect significantly more pronounced in younger patients [13].

It is important to note that more recent studies have reported that obesity may be associated with better prognosis and lower mortality in conditions such as pneumonia, ARDS, COPD, and lung cancer—a phenomenon referred to as the "obesity paradox." While this paradox remains both recognized and debated in epidemiological studies, emerging research has begun to propose potential underlying mechanisms [28].

Our study aims to further delineate the impact of obesity on mortality as well as explore its impact on more granular and longitudinal outcomes such as ICU length of stay, need for tracheostomy, and length of mechanical ventilation. By confining the scope of our study to the pre-vaccine era and excluding patients with advanced directives, cancer, or ESRD, we hope to reduce confounding factors and enhance the internal validity of our study.

## Methods

### Study design and database

We designed a retrospective cohort study using the HCA Healthcare electronic inpatient database. This database represents 149 hospitals in 18 states in the U.S. We queried the database from March 1, 2020, to December 31, 2020 to intentionally exclude patients who could have undergone COVID-19 vaccination.

### Patient population

We included all patients aged 18 years or older who were hospitalized in an HCA Healthcare hospital during the study period and had a positive test for severe acute respiratory syndrome coronavirus 2 (SARS-CoV-2) during their admission. Exclusion criteria included encounters with incomplete biographical data, incomplete or outlier BMI data, duplicate encounters, DNR/DNI status, and comorbidities considered by the authors as significantly altering the course of illness: pre-existing malignancy, chronic pulmonary disease, immunocompromised including Human Immunodeficiency Virus [HIV], autoimmune disease, kidney disease requiring dialysis, heart failure with reduced ejection fraction, cardiac congenital and valvular disorders, cor pulmonale, solid organ transplant, splenectomy, liver cirrhosis, and chronic use of immunosuppressive medications. The list of diagnoses, part of the exclusion criteria, was agreed upon by the authors as diseases with an inherited potential for an unfavorable infection course due to immune system impairment or chronic organ failure.

### Study outcomes

The primary outcome was a composite of inpatient mortality or hospice admissions among hospitalized COVID-19 patients according to their admission body mass index. Secondary outcomes included inpatient mortality, need for mechanical ventilation, need for tracheostomy, duration of mechanical ventilation, and ICU length of stay (LOS)according to their BMI.

### Variables of interest

Data extracted included patient demographics, age, gender, race, and ethnicity. We collected BMI values and classified them into four categories: underweight ($<18.5\,kg/m^2$), average ($18.5$–$24.9\,kg/m^2$), overweight ($25$–$29.9\,kg/m^2$), and obese ($\geq30\,kg/m^2$). We also collect data on patients' comorbidities, including diabetes mellitus, hypertension, chronic kidney disease, and coronary artery disease. We did not collect prior SARS-CoV-2 infection serostatus nor vaccination status (the vaccination program was not available during the study timeframe) and those variables were excluded from the analysis.

### Statistical analysis

Categorical variables were described using frequency (percentage) and analyzed for significance using the Chi-squared test. Continuous variables were described using mean (standard deviation [SD]). Continuous variables were analyzed for significance using ANOVA for parametric variables. A post-hoc pairwise comparisons test using the Bonferroni correction was used to investigate the differences. Multivariable linear regression analysis assessed the covariates' association with duration of mechanical ventilation and ICU LOS. Multivariable logistic regression analysis assessed covariates' association with mortality or discharge to hospice, need for mechanical ventilation, or need for tracheostomy. Multivariate logistic

regression results are represented as odds ratios (ORs) and their respective 95% CIs. We adjusted for key potential confounders, including age, gender, race, ethnicity, and relevant comorbidities (diabetes, hypertension, coronary artery disease, and chronic kidney disease) selected based on established clinical relevance and prior literature indicating their impact on COVID-19 outcomes. By including them as covariates in the models, we aimed to isolate the independent association between BMI and clinical outcomes. We considered a two-sided p-value of less than.05 to indicate statistical significance with a 95% confidence interval. We performed analyses using SPSS (Statistical Package for the Social Sciences) statistical software, version 1.0.0.1461 64-bit edition, and STATA 17.

### Ethical considerations

Data extraction was performed through automated interfaces. Data was de-identified and stored in a web-based password-protected shared folder only accessible to selected authors through their institutional email accounts. This study was exempt for Institutional Review Board (IRB) oversight as it was determined not to be research with human subject. It was provided approval for implementation in accordance with current regulations and institutional policy (IRB determination reference number #2022−525).

### Results

There were 38,321 unique hospital encounters (admissions) of adult patients with COVID-19 infection during the study period. Of those, 16,325 were excluded due to incomplete biographical data (n = 908), incomplete or outlier BMI data (n = 833), duplicate encounters (n = 773), DNR/DNI status, or for having a state that significantly altering the course of illness (n = 13,811). The final data set included 21,996 unique patients selected (Fig 1).

The age of the cohort was 59.3 ± 18.2 years. The cohort was primarily male (50.3%), Caucasian (63.2%), and of non-Hispanic ethnicity (70.3%). The most represented BMI categories were obese (47.5%) and overweight (29.2%). The most prevalent comorbidities were hypertension (48%) and diabetes (35.7%). Invasive mechanical ventilation was required in 6.4% of patients, with an average duration of mechanical ventilation of 10.9 ± 15.6 days. The need for tracheostomy was reported in 1% of cases. ICU length of stay was 7.4 ± 3.9 days. All-cause mortality was reported in 4.4% (Table 1).

The cohort characteristics according to their BMI category are shown in Table 2. The obese BMI category patients were significantly younger (54.9 ± 16.5 years) than overweight (61.4 ± 17.7), normal (65.3 ± 19.4), or underweight (66.5 ± 19.5) category patients. There were more women in the obese category (53.7%) than the normal (49.3%) or overweight (42.7%) category (p< 0.05). There were more African American obese category patients (20.6%) than normal (14.7%) or overweight (14.6%) categories (p<0.05). There was a lower percentage of Caucasian patients in the obese category (61.0%) when compared to underweight (68.3%), normal (67.1%) or overweight (63.6%) categories. (p<0.05). Hispanics ethnicity was more frequently recorded in the obese (27.5%) and overweight (28.4%) than the normal (20.6%) or underweight (12.2%) categories (p<0.05).

The clinical outcomes of the cohort by BMI category are shown in Table 3. There was no significant difference in the all-cause mortality and hospice admission rate across BMI categories. Obese patients had a higher need for mechanical ventilation (7.8%) than the overweight (5.8%), normal (4.6%) or underweight (3.9%) categories (p<0.05). There was a higher tracheostomy rate in patients in the obese category (1.2%) than in the normal category (0.6%) patients (p<0.05). The duration of mechanical ventilation was higher in overweight (12.1 ± 19.1) or obese (11.5 ± 15.1) category patients when compared to normal (6.8 ± 8.7) category patients. (p<0.05). ICU length of stay was higher in the obese (8.3 ± 10.4) and overweight (7.7 ± 10.9) category patients when compared to normal (5.1 ± 6.4) or underweight (4.4 ± 5.6) category patients (p<0.05) (Fig 2).

All-cause mortality and hospice admissions was independently predicted by age (OR 1.03, 95% CI, 1.02 to 1.03, p<0.001), male gender (OR 1.53, 95% CI, 1.34 to 1.74, p<0.001), Hispanic ethnicity (OR 1.65, 95% CI, 1.39 to 1.94,

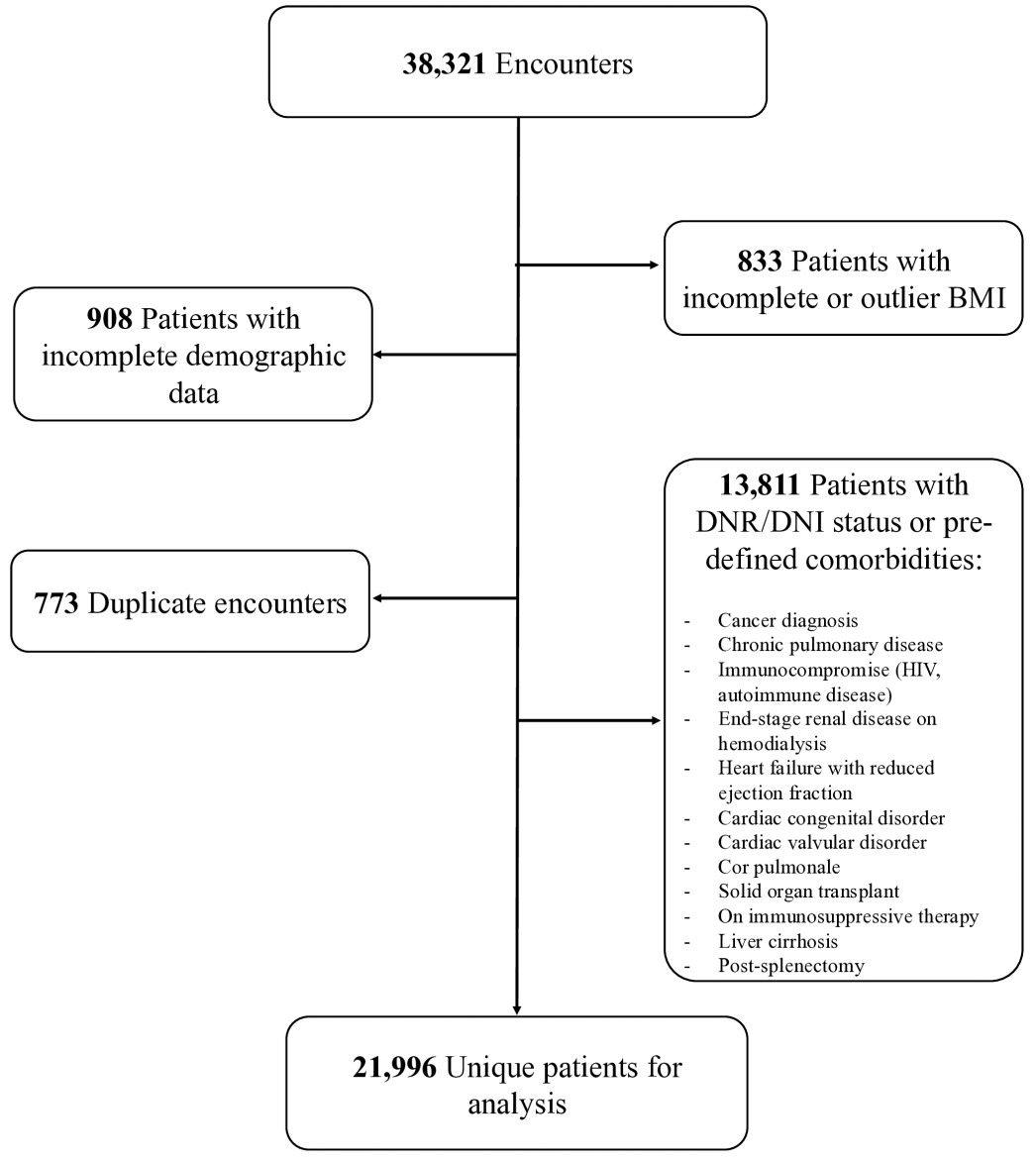

<sup>a</sup> Patients selected on first hospital admission

BMI: Body mass index; HIV: Human Immunodeficiency Virus; DNR: Do not resuscitate; DNI: Do not intubate

**Fig 1. Study population selection.**

p<0.001), diabetes (OR 1,47, 95% CI, 1.28 to 1.69, p < 0.001), hypertension (OR 0.83, 95% CI, 0.72 to 0.97, p < 0.05), chronic kidney disease (OR 1.30, 95% CI, 1.07 to 1..58, p < 0.05) and obese BMI category (OR 1.29, 95% CI, 1.08 to 1.55, p<0.05) (Table 4, Fig 3).

Inpatient mortality was independently predicted by age (OR 1.01, 95% CI, 1.01 to 1.02, p<0.001), male gender (OR 1.78, 95% CI, 1.52 to 2.08, p<0.001), African American race (OR 1.34, 95% CI, 1.08 to 1.65, p<0.05), Hispanic ethnicity (OR 1.92, 95% CI, 1.59 to 2.32, p<0.001), diabetes (OR 1.54, 95% CI, 1.31 to 1.81, p<0.001), chronic kidney disease (OR 1.49, 95% CI, 1.19 to 1..86, p<0.001) and obese BMI category (OR 1.67, 95% CI, 1.34 to 2.08, p<0.001) (Table 5, Fig 4).

**Table 1. Cohort characteristics.**

| Characteristic | Cohort (n = 21,996) |
|---|---|
| Age, years, mean ± SD | 59.3 ± 18.2 |
| Gender, n (%) | |
| Female | 10,932 (49.7) |
| Male | 11,064 (50.3) |
| Race, n (%) | |
| African American | 4,223 (19.2) |
| Caucasian | 13,898 (63.2) |
| Other | 3,875 (17.6) |
| Ethnicity, n (%) | |
| Hispanic | 5,711 (26.0) |
| Non-Hispanic | 15470 (70.3) |
| Not specified | 815 (3.7) |
| BMI, kg/m$^2$, mean ± SD | 30.8 ± 29.5 |
| BMI Category, n (%) | |
| Underweight (<18.5 kg/m$^2$) | 565 (2.6) |
| Normal (18.5–24.9 kg/m$^2$) | 4,495 (20.4) |
| Overweight (25–29.9 kg/m$^2$) | 6,495 (29.2) |
| Obese (≥30 kg/m$^2$) | 10,441 (47.5) |
| Comorbidities, n (%) | |
| Diabetes | 7,845 (35.7) |
| Hypertension | 10,562 (48.0) |
| Chronic Kidney Disease | 2,670 (12.1) |
| Coronary Artery Disease | 3,300 (15.0) |
| Need for mechanical ventilation, n (%) | 1,417 (6.4) |
| Duration of mechanical ventilation, days, mean ± SD | 10.9 ± 15.6 |
| Need for tracheostomy placement, n (%) | 219 (1.0) |
| ICU length of stay, days, mean ± SD | 7.4 ± 3.9 |
| All-cause mortality or hospice admission, n (%)[a] | 990 (4.5) |

[a]Includes mortality that occurred inpatient and in hospice services.

SD = Standard Deviation. BMI = Body Mass Index. ICU = Intensive care unit.

The need for mechanical ventilation was independently predicted by age (OR 0.99, 95% CI, 0.98-0.99), male gender (OR 1.50, 95% CI, 1.34 to 1.68, p<0.001), Hispanic ethnicity (OR 1.21, 95% CI, 1.06 to 1.39, p<0.05), not-specified ethnicity (OR 1.56, 95% CI, 1.20 to2.03, p<0.001), diabetes (OR 1.57, 95% CI 1.39 to 1.77, p<0.001), hypertension (OR 1.14, 95% CI, 1.00 to 1.29, p<0.05), chronic kidney disease (OR 1.43, 95% CI, 1.19 to 1.71, p<0.001) and obese BMI category (OR 1.55, 95% CI, 1.31 to 1.82, p<0.001) (Table 6, Fig 5).

The need for tracheostomy placement was independently predicted by age (OR 0.98, 95% CI, 0.97 to 0.98, p<0.001), diabetes (OR 1.70, 95% CI 1.28 to 2.27, p<0.001), hypertension (OR 1.75, 95% CI, 1.29 to 2.39, p<0.001), chronic kidney disease (OR 1.87, 95% CI, 1.19 to 2.95, p<0.05) and obese BMI category (OR 1.57, 95% CI, 1.03 to 2.41, p<0.05) (Table 7, Fig 6).

The duration of mechanical ventilation was predicted by Age (β −0.12, 95% CI, −0.18 to −0.06, p < 0.001), race other than Caucasian or African American (β 2.47, 95% CI, 0.32 to 4.62, p<0.05), hypertension (β 2.01, 95% CI, 0.23 to 3.78, p<0.05), coronary artery disease (β −2.35, 95% CI, −4.56 to −0.13, p<0.05), overweight (β 4.8, 95% CI, 2.19 to 7.41, p<0.001), and obesity (β 3.67, 95% CI, 1.28 to −6.06, p<0.001) (Table 8).

**Table 2. Cohort Demographics by BMI Category.**

| BMI Category | | | | | |
|---|---|---|---|---|---|
| Characteristic | Underweight | Normal | Overweight | Obese | p-value |
| Age, years, mean ± SD | 66.5 ± 19.5[a] | 65.3 ± 19.4[a] | 61.4 ± 17.7[b] | 54.9 ± 16.5[c] | < 0.001 |
| Gender, n (%) | | | | | < 0.001 |
| Female | 335 (59.3)[b] | 2217 (49.3)[a] | 2776 (42.7)[c] | 5604 (53.7)[b] | |
| Male | 230 (40.7)[b] | 2278 (50.7)[a] | 3719 (57.3)[c] | 4837 (46.3)[b] | |
| Race, n (%) | | | | | < 0.001 |
| African American | 112 (19.8)[b] | 661 (14,7)[a] | 948 (14.6)[a] | 2154 (20.6)[b] | |
| Caucasian | 386 (68.3)[a,b] | 3017 (67.1)[a] | 4130 (63.6)[b] | 6365 (61.0)[c] | |
| Other | 67 (11.9)[b] | 817 (18.2)[a] | 1417 (21.8)[c] | 1922 (18.4)[a] | |
| Ethnicity, n (%) | | | | | < 0.001 |
| Hispanic | 69 (12.2)[b] | 924 (20.6)[a] | 1842 (28.4)[c] | 2876 (27.5)[c] | |
| Non-Hispanic | 381 (67.4)[b] | 3419 (76.1)[a] | 4442 (68.4)[b] | 7228 (69.2)[b] | |
| Not specified | 115 (20.4)[b] | 152 (3.4)[a] | 211 (3.2)[a] | 337 (3.2)[a] | |

Values in the same row and subtable not sharing the same subscript are significantly different at p < 0.05. Cells with no subscript are not included in the analysis. Test assume equal variances.

Test are adjusted for all pairwise comparisons within a row of each innermost subtable using the Bonferroni correction. SD = Standard Deviation. BMI = Body Mass Index. ICU = Intensive care unit.

All p values below 0.001 were expressed as p < 0.001.

**Table 3. Clinical Outcomes by BMI Category.**

| BMI Category | | | | | |
|---|---|---|---|---|---|
| Characteristic | Underwe ight | Normal | Overwei ght | Obese | pvalue |
| All-cause mortality or hospice admission, n (%) | 32 (5.7)[a] | 205 (4.6)[a] | 264 (4.1)[a] | 489 (4.7)[a] | 0.14 |
| Need for mechanical ventilation, n (%) | 22 (3.9)a,b | 206 (4.6)[a] | 379 (5.8)[b] | 810 (7.8)[c] | < 0.001 |
| Need for tracheostomy placement, n (%) | 1 (0.2)a,b | 26 (0.6)[a] | 64 (1.0)a,b | 128 (1.2)[b] | < 0.001 |
| Duration of mechanical ventilation, days, mean ± SD | 5.4 ± 7.6a,b | 6.8 ± 8.7[a] | 12.1 ± 19.1[b] | 11.5 ± 15.1b,c | < 0.001 |
| ICU length of stay, days, mean ± SD | 4.4 ± 5.6[a] | 5.1 ± 6.4[a] | 7.7 ± 10.9[b] | 8.3 ± 10.4[b] | < 0.001 |

Values in the same row and subtable not sharing the same subscript are significantly different at p < 0.05. Cells with no subscript are not included in the analysis. Test assume equal variances.

Test are adjusted for all pairwise comparisons within a row of each innermost subtable using the Bonferroni correction.

SD = Standard Deviation. BMI = Body Mass Index. ICU = Intensive care unit All p values below 0.001 were expressed as p < 0.001.

## Discussion

Our study found that obese patients have higher odds of all-cause mortality and hospice admission, in-patient mortality, invasive mechanical ventilation use, and tracheostomy placement when compared to other BMI groups. Critically ill obese patients had a longer duration of invasive mechanical ventilation and ICU length of stay.

The overall mortality and hospice admission rate in our cohort was reported at 4.5%, which falls within the lower quartile when compared to other studies during pre-vaccination times (3.2% to 60.9%) [19,29]. This finding can be attributed to the enhanced access to care and timely interventions compared to other settings with limited resources. Obesity has been associated with increased severity and mortality due to COVID-19 infection [10–12,30], while being overweight increases the risk of hospitalizations but not mortality [10,31] in some studies.

In contrast, others do find a link with survival [19,27,32].

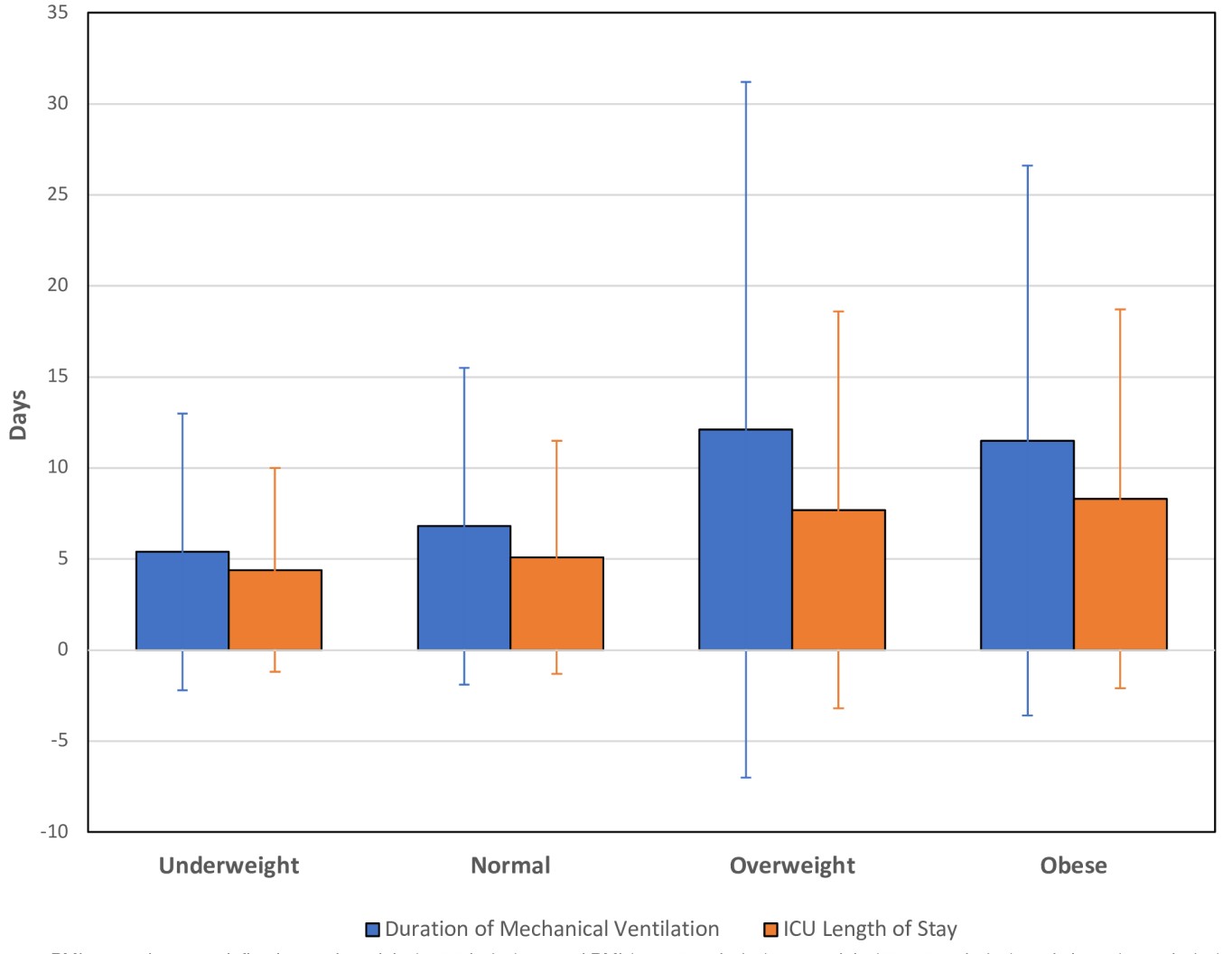

BMI categories were defined as underweight (<18.5 kg/m²), normal BMI (18.5–24.9 kg/m²), overweight (25.0–29.9 kg/m²), and obese (≥30.0 kg/m²)

**Fig 2. Days of Mechanical Ventilation and ICU Stay by BMI Category.**

In our study, all-cause mortality and admission to hospice were similar across all BMI categories. Further analysis detected that obese patients were 1.29 times more likely to die or be admitted to hospice, and 1.67 times more likely to die during hospitalization when compared to patients with a normal BMI. While some of these odds ratios (ORs) may be considered modest in magnitude, with limited individual-level impact, the consistency of these associations, which include diverse geographic regions as a national dataset, combined with their population-level implications, underscores their clinical relevance. Even a modest increase in risk, when applied to a large population, particularly during a global pandemic, can result in substantial added strain on the healthcare system. Our data supports previously significant surges in hospice admissions in the obese population when comparing pre- and post-COVID-19 outbreak periods (23.3% vs 52.8%, p<0.001) [33,34] and although initial adjusted composite outcome showed no differences among BMI category, the regression models accounting for other cofounders revealed an association that strengthen when evaluating mortality only. It is possible that hospice admissions might have been driven by other factors not accounted in our initial analysis diluting composite outcome differences between BMI groups.

**Table 4. Multivariate Logistic Regression Analysis examining Predictors of All–Cause Mortality or Hospice Admission.**

| Variable | OR | 95% CI | p-value |
|---|---|---|---|
| Age (years) | 1.03 | 1.02 to 1.03 | < 0.001 |
| Gender (ref. female) | | | |
| Male | 1.53 | 1.34 to 1.74 | < 0.001 |
| Race (ref. Caucasian) | | | |
| African American | 1.14 | 0.94 to 1.37 | 0.18 |
| Other | 0.92 | 0.76 to 1.11 | 0.37 |
| Ethnicity (ref, non-Hispanic) | | | |
| Hispanic | 1.65 | 1.39 to 1.94 | < 0.001 |
| Not specified | 1.06 | 0.74 to 1.52 | 0.75 |
| Diabetes (ref. no) | 1.47 | 1.28 to 1.69 | < 0.001 |
| Hypertension (ref. no) | 0.83 | 0.72 to 0.97 | < 0.05 |
| Chronic Kidney Disease (ref. no) | 1.3 | 1.07 to 1.58 | < 0.05 |
| Coronary Artery Disease (ref. no) | 1.13 | 0.96 to 1.33 | 0.14 |
| BMI Category (ref. normal) | | | |
| Underweight (<18.5 kg/m²) | 1.34 | 0.91 to 1.99 | 0.14 |
| Overweight (25–29.9 kg/m²) | 0.91 | 0.75 to 1.10 | 0.34 |
| Obese (≥30 kg/m²) | 1.29 | 1.08 to 1.55 | < 0.05 |

Analysis performed in 21996 encounters.

BMI = Body Mass Index. CI = Confidence Interval.

All p-values below 0.001 were expressed as p < 0.001. All p-values below 0.05 were expressed as p < 0.05.

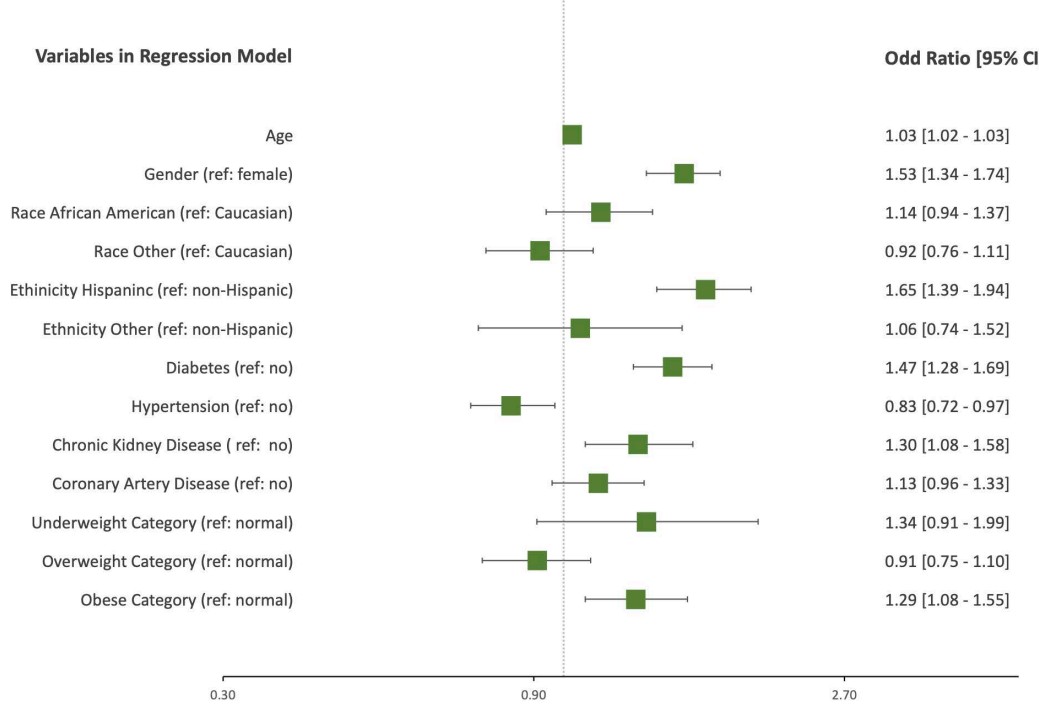

BMI categories were defined as underweight (<18.5 kg/m²), normal BMI (18.5–24.9 kg/m²), overweight (25.0–29.9 kg/m²), and obese (≥30.0 kg/m²)

**Fig 3. Multivariate Logistic Regression Analysis examining Predictors of All-CauseMortality or Hospice Admission.**

**Table 5. Multivariate Logistic Regression Analysis Examining Predictors of Inpatient Mortality.**

| Variable | OR | 95% CI | p-value |
|---|---|---|---|
| Age (years) | 1.01 | 1.01 to 1.02 | < 0.001 |
| Gender (ref. female) | | | |
| Male | 1.78 | 1.52 to 2.08 | < 0.001 |
| Race (ref. Caucasian) | | | |
| African American | 1.34 | 1.08 to 1.65 | <0.05 |
| Other | 1.04 | 0.85 to 1.28 | 0.68 |
| Ethnicity (ref, non-Hispanic) | | | |
| Hispanic | 1.92 | 1.59 to 2.32 | < 0.001 |
| Not specified | 1.12 | 0.72 to 1.73 | 0.61 |
| Diabetes (ref. no) | 1.54 | 1.31 to 1.81 | < 0.001 |
| Hypertension (ref. no) | 0.93 | 0.78 to 1.11 | 0.45 |
| Chronic Kidney Disease (ref. no) | 1.49 | 1.19 to 1.86 | <0.001 |
| Coronary Artery Disease (ref. no) | 1.18 | 0.97 to 1.43 | 0.09 |
| BMI Category (ref. normal) | | | |
| Underweight (<18.5 kg/m²) | 0.85 | 0.46 to 1.59 | 0.63 |
| Overweight (25–29.9 kg/m²) | 1.05 | 0.83 to 1.34 | 0.66 |
| Obese (≥30 kg/m²) | 1.67 | 1.34 to 2.08 | < 0.001 |

Analysis performed in 21996 encounters.

BMI = Body Mass Index. CI = Confidence Interval.

All p-values below 0.001 were expressed as p < 0.001. All p-values below 0.05 were expressed as p < 0.05.

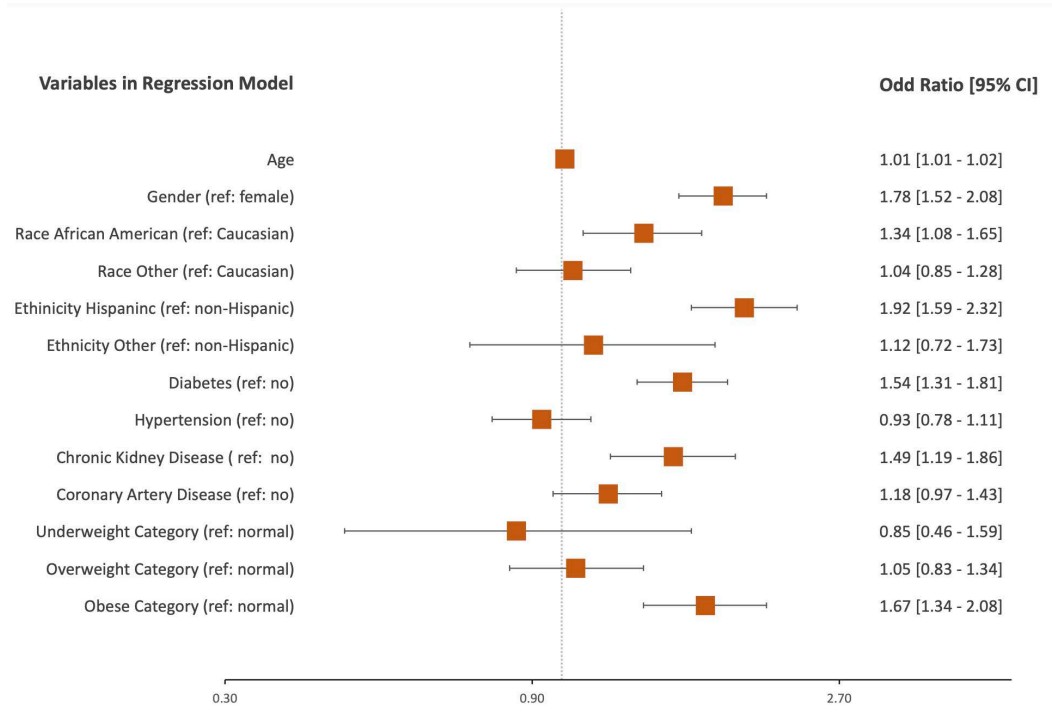

BMI categories were defined as underweight (<18.5 kg/m²), normal BMI (18.5–24.9 kg/m²), overweight (25.0–29.9 kg/m²), and obese (≥30.0 kg/m²)

**Fig 4. Multivariate Logistic Regression Analysis Examining Predictors of Inpatient Mortality.**

**Table 6. Multivariate Logistic Regression Analysis Examining Predictors of Need for Invasive Mechanical Ventilation.**

| Variable | OR | 95% CI | p-value |
|---|---|---|---|
| Age (years) | 0.99 | 0.98 to 0.99 | < 0.001 |
| Gender (ref. female) | | | |
| Male | 1.50 | 1.34 to 1.68 | < 0.001 |
| Race (ref. Caucasian) | | | |
| African American | 0.88 | 0.75 to 1.03 | 0.11 |
| Other | 1.01 | 0.87 to 1.18 | 0.87 |
| Ethnicity (ref, non-Hispanic) | | | |
| Hispanic | 1.21 | 1.06 to 1.39 | < 0.05 |
| Not specified | 1.56 | 1.20 to 2.03 | < 0.001 |
| Diabetes (ref. no) | 1.57 | 1.39 to 1.77 | < 0.001 |
| Hypertension (ref. no) | 1.14 | 1.00 to 1.29 | < 0.05 |
| Chronic Kidney Disease (ref. no) | 1.43 | 1.19 to 1.71 | < 0.001 |
| Coronary Artery Disease (ref. no) | 1.16 | 0.99 to 1.35 | 0.06 |
| BMI Category (ref. normal) | | | |
| Underweight (<18.5 kg/m²) | 0.84 | 0.53 to 1.32 | 0.45 |
| Overweight (25–29.9 kg/m²) | 1.16 | 0.98 to 1.39 | 0.09 |
| Obese (≥30 kg/m²) | 1.55 | 1.31 to 1.82 | < 0.001 |

Analysis performed in 21996 encounters.

BMI = Body Mass Index. CI = Confidence Interval.

All p values below 0.001 were expressed as p < 0.001. All p values below 0.05 were expressed as p < 0.05.

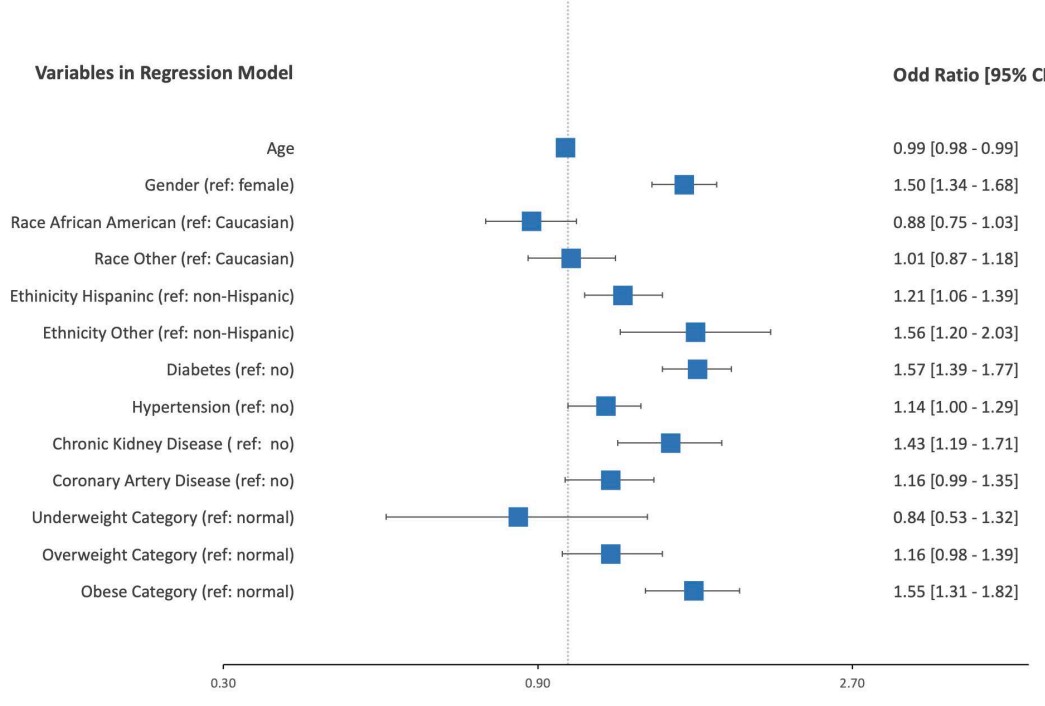

BMI categories were defined as underweight (<18.5 kg/m²), normal BMI (18.5–24.9 kg/m²), overweight (25.0–29.9 kg/m²), and obese (≥30.0 kg/m²)

**Fig 5. Multivariate Logistic Regression Analysis Examining Predictors of Need for Invasive Mechanical Ventilation.**

**Table 7. Multivariate Logistic Regression Analysis Examining Predictors of Need for Tracheostomy Placement.**

| Variable | OR | 95% CI | p-value |
|---|---|---|---|
| Age (years) | 0.98 | 0.97 to 0.98 | < 0.001 |
| Gender (ref. female) | | | |
| Male | 1.29 | 0.99 to 1.69 | 0.06 |
| Race (ref. Caucasian) | | | |
| African American | 0.72 | 0.48 to 1.06 | 0.09 |
| Other | 1.33 | 0.94 to 1.88 | 0.11 |
| Ethnicity (ref, non-Hispanic) | | | |
| Hispanic | 0.88 | 0.63 to 1.24 | 0.46 |
| Not specified | 1.14 | 0.57 to 2.28 | 0.7 |
| Diabetes (ref. no) | 1.7 | 1.28 to 2.27 | < 0.001 |
| Hypertension (ref. no) | 1.75 | 1.29 to 2.39 | < 0.001 |
| Chronic Kidney Disease (ref. no) | 1.87 | 1.19 to 2.95 | < 0.05 |
| Coronary Artery Disease (ref. no) | 0.78 | 0.50 to 1.19 | 0.25 |
| BMI Category (ref. normal) | | | |
| Underweight (<18.5 kg/m²) | 0.47 | 0.09 to 2.41 | 0.37 |
| Overweight (25–29.9 kg/m²) | 1.45 | 0.92 to 2.27 | 0.11 |
| Obese (≥30 kg/m²) | 1.57 | 1.03 to 2.41 | < 0.05 |

Analysis performed in 21996 encounters.

BMI = Body Mass Index. CI = Confidence Interval.

All p values below 0.001 were expressed as p < 0.001. All p values below 0.05 were expressed as p < 0.05.

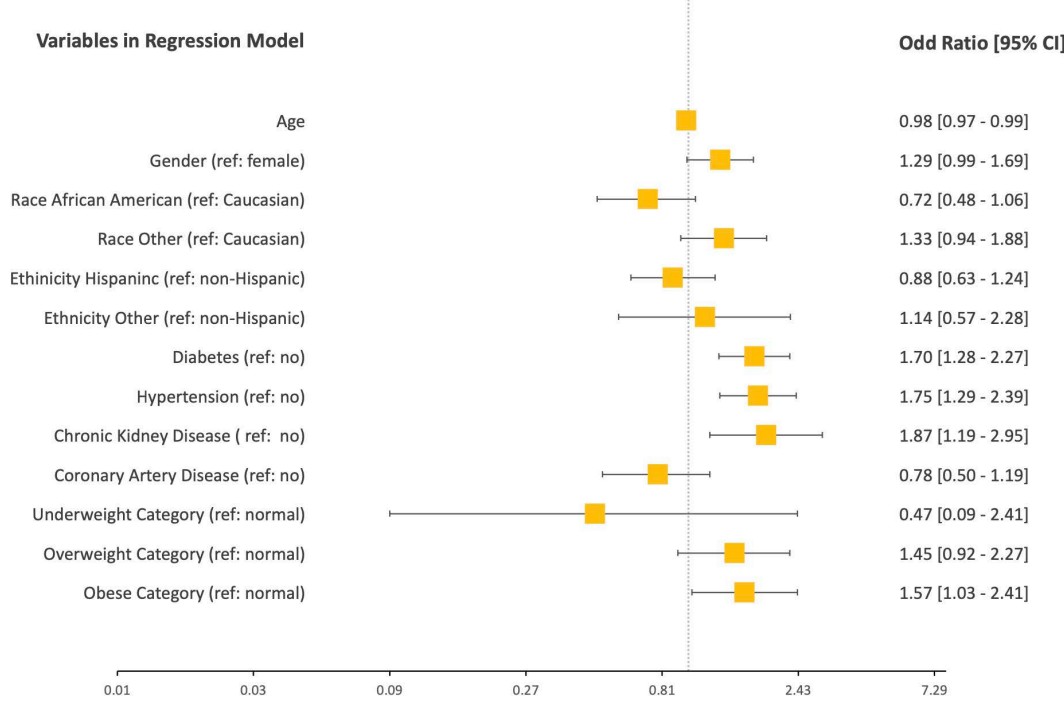

BMI categories were defined as underweight (<18.5 kg/m²), normal BMI (18.5–24.9 kg/m²), overweight (25.0–29.9 kg/m²), and obese (≥30.0 kg/m²)

**Fig 6. Multivariate Logistic Regression Analysis Examining Predictors of Need for Tracheostomy Placement.**

**Table 8. Multivariate Linear Regression Analysis on Determinants of Duration of Mechanical Ventilation.**

| Variable | β | 95% CI | p-value |
|---|---|---|---|
| Age (years) | −0.12 | −0.18 to −0.06 | < 0.001 |
| Gender (ref. female) | | | |
| Male | 0.46 | −1.23 to 2.08 | 0.61 |
| Race (ref. Caucasian) | | | |
| African American | −0.36 | −2.68 to 1.97 | 0.76 |
| Other | 2.47 | −0.31 to 4.62 | < 0.05 |
| Ethnicity (ref, non-Hispanic) | | | |
| Hispanic | 1.61 | −0.37 to 3.59 | 0.11 |
| Not specified | 0.48 | −3.30 to 4.27 | 0.25 |
| Diabetes (ref. no) | 1.47 | −0.22 to 3.17 | 0.09 |
| Hypertension (ref. no) | 2.01 | 0.23 to 3.78 | < 0.05 |
| Chronic Kidney Disease (ref. no) | 0.22 | −2.29 to 2.73 | 0.86 |
| Coronary Artery Disease (ref. no) | −2.35 | −4.56 to −0.13 | < 0.05 |
| BMI Category (ref. normal) | | | |
| Underweight (<18.5 kg/m²) | −0.53 | −7.32 to 6.29 | 0.88 |
| Overweight (25–29.9 kg/m²) | 4.80 | 2.19 to 7.41 | < 0.001 |
| Obese (≥30 kg/m²) | 3.67 | 1.28 to 6.06 | < 0.001 |

Analysis performed in 1417 encounters. $R^2 = 0.053$.

BMI = Body Mass Index. CI = Confidence Interval.

All p values below 0.001 were expressed as $p < 0.001$. All p values below 0.05 were expressed as $p < 0.05$.

ICU length of stay was predicted by male gender (β 0.98, 95% CI, 0.39 to 1.58, $p < 0.05$), race other than Caucasian or African American (β 1.29, 95% CI, 0.47 to 2.11, $p < 0.05$), Hispanic ethnicity (β 1.35, 95% CI, 0.57 to 2.13, $p < 0.001$), diabetes (β 0.94, 95% CI, 0.32 to 1.56, $p < 0.05$), coronary artery disease (β -1.23, 95% CI, -2.04 to -0.40, $p < 0.05$), overweight (β 2.2, 414 95% CI, 1.34 to 3.10, $p < 0.001$), and obesity (β 2.90, 95% CI, 2.08 to 3.72, $p < 0.001$) (Table 9).

**Table 9. Multivariate linear regression analysis on determinants of ICU length of stay.**

| Variable | β | 95% CI | p-value |
|---|---|---|---|
| Age (years) | −0.02 | −0.04 to 0.00 | 0.13 |
| Gender (ref. female) | | | |
| Male | 0.98 | 0.39 to 1.58 | < 0.05 |
| Race (ref. Caucasian) | | | |
| African American | −0.45 | −1.28 to 0.39 | 0.29 |
| Other | 1.29 | 0.47 to 2.11 | < 0.05 |
| Ethnicity (ref, non-Hispanic) | | | |
| Hispanic | 1.35 | 0.57 to 2.13 | < 0.001 |
| Not specified | 0.18 | −1.22 to 1.59 | 0.79 |
| Diabetes (ref. no) | 0.94 | 0.32 to 1.56 | < 0.05 |
| Hypertension (ref. no) | 0.54 | −0.11 to 1.19 | 0.11 |
| Chronic Kidney Disease (ref. no) | 0.02 | −0.92 to 0.97 | 0.95 |
| Coronary Artery Disease (ref. no) | −1.23 | −2.04 to 0.40 | < 0.05 |
| BMI Category (ref. normal) | | | |
| Underweight (<18.5 kg/m²) | −0.24 | −2.15 to 1.69 | 0.80 |
| Overweight (25–29.9 kg/m²) | 2.22 | 1.34 to 3.10 | < 0.001 |
| Obese (≥30 kg/m²) | 2.90 | 2.08 to 3.72 | < 0.001 |

Analysis performed in 4288 encounters. $R^2 = 0.037$.

BMI = Body Mass Index. CI = Confidence Interval.

All p values below 0.001 were expressed as $p < 0.001$. All p values below 0.05 were expressed as $p < 0.05$.

Obese patients were found in almost half of our cohort of hospitalized patients with COVID-19-related illness. Such findings support the severity of the obesity pandemic in the US population. In the past two decades, the obese population has risen from 30.5% to 41.9%. The severely obese population (BMI of > 40.0kg/m2) has doubled from 4.7% to 9.2%. The higher percentage of obesity in the general population is a likely contributor to the increased propensity of this BMI group to be admitted to the hospital with COVID-19-related illness [27]. Weight management and obesity prevention strategies become cornerstone interventions to mitigate the severity of COVID-19-related illness outcomes. Given the observed associations between higher BMI and adverse clinical trajectories, our findings support broader public health initiatives aimed at addressing obesity as a modifiable risk factor in pandemic preparedness and response efforts.

The obese patients who were admitted to the hospital with COVID-19-related illness were approximately ten years younger and had a disproportionately higher presence of African American women when compared to those patients with low and normal BMI, and six years younger than the overweight cohort. Additionally, Hispanic representation was higher in the obese and overweight BMI categories than in the normal or underweight BMI categories [24,31].

While prior studies identified male sex and non-Hispanic African American adults as having higher hospitalization risk [24], our findings show that male sex and Hispanic ethnicity were independently associated with increased all-cause mortality, hospice admission, inpatient mortality, mechanical ventilation, and longer ICU stays. African American race was also significantly associated with higher inpatient mortality.

The higher risk observed in males may be partly explained by biological factors. Testosterone exerts immunosuppressive effects, while estrogen enhances immune response and may offer protection by reducing ACE-2 receptor expression, potentially limiting viral entry and disease severity [35,36]. Racial and ethnic disparities likely reflect systemic inequities, including limited healthcare access, greater prevalence of chronic conditions, and social determinants such as income, housing, and occupation [37–39]. These findings highlight the urgent need to address health inequities and expand access to care.

Several factors in obese individuals enhance the severity of COVID-19-related illnesses. Severe acute respiratory virus 2 (SARS-CoV-2) attaches to the ACE-2 receptor, which is tightly regulated by the renin-angiotensin-aldosterone system (RAAS), which is imbalanced in obese patients [7,16,19]. Adipose cells contain ACE-2 receptors and an ample supply of inflammatory molecules, such as Interleukin-4 (IL-4), which ultimately may contribute to a more robust inflammatory response leading to severe disease [40]. In obesity, leukocyte responses to inflammation are amplified by adipose-derived signals. Adipocytes secrete pro-inflammatory adipokines like leptin and express higher levels of ACE-2, MCP-1, and IL-4, creating a primed immune state. This chronic inflammatory milieu can contribute to a subsequent "metabolic reprogramming" of innate immune cells, particularly neutrophils and macrophages, causing over sensitization and amplified inflammatory signaling in response to viral stimuli, predisposing an obese individual to cytokine storm. Obese individuals are also prone to heightened macrophage activation, contributing to lung inflammation, ARDS, and increased COVID-19 mortality, linking altered immunometabolism to worse respiratory outcomes [16,18,21,41]. Furthermore, obese patients have a particular propensity for thrombus formation and endothelial adhesion due to the excessive production of adhesion molecules (adipocytokines, thrombospondin-1, and TNF-α), promoting the synthesis of plasminogen activation inhibitor 1 (PAI-1 [42]. Adipokine imbalance has been associated with a higher risk of ARDS and infections, particularly pneumonia [43–47].

Lastly, elevated monocyte chemoattractant protein 1 (MCP-1) levels in obese individuals contribute to immune cell recruitment and inflammation in adipose tissue and arterial walls, heightening cardiovascular risks [42,48,49], and promoting the development of chronic comorbidities (Type 2 diabetes, coronary artery disease, and hypertension) [48–50].

Our study revealed both higher utilization and longer duration of respiratory support in the ICU. COVID-19-related illness patients requiring IMV (Invasive Mechanical Ventilation) increase as BMI increases, with up to 85.7% of patients with a BMI >35 kg m$^2$ needing IMV upon admission [51]. After adjusting for age, sex, diabetes, and hypertension, severe obesity (BMI >35kg/ m2) remained independently associated with a greater than sevenfold increase in the odds

of requiring IMV compared to individuals with normal body weight [51]. Our study showed that obese patients were 1.5 times more likely to require mechanical ventilation compared to patients with a normal BMI. The average duration of mechanical ventilation for overweight and obese patients was also longer (3-5 days) compared to patients with a normal BMI.

Liberating obese patients from mechanical ventilation can be challenging due to changes in respiratory mechanics, such as restrictive physiology from the heavy chest wall and impaired diaphragm excursion, decreasing lung volumes [52,53] and functional residual capacity (FRC) which decreases approximately by 5-15% for every 5 kg/m² increase in BMI [54] promoting airway closure and atelectasis [52]. This could be exacerbated due to sedatives commonly used in ICU settings in ventilated patients, decreasing muscle tone, further impairing the respiratory function [52,55]. Adipose tissue accumulation in the abdominal and chest wall causes increased intra-abdominal and intra-thoracic pressures, worsening respiratory mechanics, and respiratory disease severity in obese patients with COVID-19 infection [19,56–58]. The obese patient also carries an increased incidence of related respiratory pathology, including sleep-disordered breathing (SDB), chronic hypercapnic respiratory failure, and asthma (starting in overweight individuals and showing a dose-response effect with increasing risk as BMI increases) [59,60] This associated pathology could hinder a positive respiratory clinical outcome and related to higher susceptibility to prolonged and more severe COVID-19-induced ARDS, increasing the likelihood of developing chronic respiratory failure and need for tracheostomy placement [61] as confirmed in our study. An interesting point is the fact we found a inverse association between age and need for mechanical ventilation and it duration as well as for tracheostomy placement. The authors infer that advance age likely influence the decision of clinical in placement patients on IMV and to proceed with tracheostomy leaning towards a less invasive approach and more emphasis on goals of care discussions.

Overweight and obese patients may spend an additional 2 to 3-day-period in the ICU compared to patients with normal BMI, suggesting a more protracted recovery process [20]. Obese patients have substantially longer hospital and ICU stays, averaging 4 days longer than their underweight counterparts.. As BMI increases, so does the length of ICU stay, highlighting the influence of weight status on healthcare resource utilization in critical care settings [20,49].

Obese patients with COVID-19 infection have also been associated with delays in seeking medical attention, contributing to a more severe disease trajectory and ultimately to more ICU admissions and increased strain on hospital resources [33] and costs [20].

Our study has several limitations: 1) Our selected criteria were limited to patients admitted to the hospital with a positive test for COVID-19 infection. Although most of the disease is expected to be related to respiratory pathology, we could not exclude patients who were hospitalized due to non-respiratory COVID-19 related illness nor account for the possibility of co-infection with other viral pathogens with symptoms that mimic COVID-19 infection through point-of-care testing [62,63]; 2) We relied on height and weight in medical records to categorize patient by BMI which brings challenges such as using an estimated height measured in the supine position, not accounting for fluid shifts experiences by the critically ill due to resuscitation maneuvers and extended ICU stay, the positive fluid balance related to interventions and differences in actual weight versus dry weight in patient with new needs for hemodialysis or worsening heart failure or differences in muscle mass; 3) Our data collection timeframe occurred before the availability of target vaccination yet temporal bias could have been introduce due to the evolving nature of treatment protocols (including anti-inflammatory and antiviral therapies) that emerge during the first stages of the pandemic. 4) Our study did not distinguish timing to mechanical ventilation or to tracheostomy that might have been skewed in the obese patient due to perceived anatomic and physiologic challenges by clinicians; 5) We did not include patients with other comorbidities, such as immunocompromised states, or advanced directives (do not resuscitate, do not intubated) that might have impacted clinical outcomes by introducing survivorship bias; 6) Our predictor models did account for clinical severity scores timing of interventions a data set limitation nor interaction between common obesity-associated comorbidities that could have impacted, clinical decision making, ICU management and clinical study outcomes.

In conclusion, this study outlines the association between obesity and adverse clinical outcomes in patients with COVID-19-related illnesses. Obesity was found to be associated with higher mortality, length of stay, and mechanical ventilation days during the pre-vaccination era of the COVID-19 pandemic. Careful respiratory vigilance, early decision-making protocols for tracheostomy when indicated, and prioritizing vaccination and early interventions are warranted in treating obese patient with COVID-19 infection in the critical care setting due to their increased risk of death and need for acute and chronic invasive ventilatory management.

## Acknowledgments

The authors thank the HCA Healthcare Florida Graduate Medical Education Research Department for their administrative support in executing this study.

This research was supported by HCA Healthcare and/or HCA Healthcare affiliated entity. The views expressed in this publication represent those of the authors and do not necessarily represent the official views of HCA Healthcare or any of its affiliated entities. The authors listed above certify that they have no affiliations with or involvement in any organization or entity with any financial or non-financial interest in the subject matter or materials discussed in this manuscript.

## Author contributions

**Conceptualization:** Andrea Folds, Mauricio Danckers.

**Data curation:** Mauricio Danckers.

**Formal analysis:** Daniel High, Mauricio Danckers.

**Investigation:** Mauricio Danckers.

**Methodology:** Andrea Folds, Mauricio Danckers.

**Project administration:** Yolanda Bonilla, Daniel High, Andrea Folds, Siddarth Kathuria, Mauricio Danckers.

**Supervision:** Mauricio Danckers.

**Validation:** Mauricio Danckers.

**Visualization:** Daniel High, Mauricio Danckers.

**Writing – original draft:** Yolanda Bonilla, Daniel High, Andrea Folds, Jose Acosta Rullan, Ana Martinez, Prerna Sharma, Mauricio Danckers.

**Writing – review & editing:** Yolanda Bonilla, Daniel High, Andrea Folds, Jose Acosta Rullan, Ana Martinez, Jude Tabba, Tanner Noris, Richard Shalmiyev, Raiko Diaz, Daniel Heller, Prerna Sharma, Siddarth Kathuria, Mauricio Danckers.

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
