## [Decision Letter · Decision Letter 0]

22 May 2025

Dear Dr. Bonilla,

Thank you for submitting your manuscript to PLOS ONE. After careful consideration, we feel that it has merit but does not fully meet PLOS ONE’s publication criteria as it currently stands. Therefore, we invite you to submit a revised version of the manuscript that addresses the points raised during the review process.

**ACADEMIC EDITOR'S COMMENTS:**

Line 142-144: "Most COVID-19-related deaths were due to progression to acute respiratory distress syndrome (ARDS) and its heightened inflammatory cytokines release state (8, 9).": this statement is inaccurate. While the original COVID due to wild-type SARS-CoV-2 strains can cause ARDS, the SARS-CoV-2 viruses that have evolved to Omicron strains primarily cause mild symptoms. The authors should introduce this point based on more references, with this one (Liu BM, et al. Genetic Conservation and Diversity of SARS-CoV-2 Envelope Gene Across Variants of Concern. J Med Virol. 2025 Jan;97(1):e70136. doi: 10.1002/jmv.70136. PMID: 39744807.) as an example (citing is optional). 

The authors should discuss a limitation that they could not exclude patients who were hospitalized due to COVID and co-infection with other non-SARS-CoV-2 respiratory pathogens. Different respiratory viruses can lead to similar symptoms with COVID, which cannot be differentiated unless performing related testing using multiplex respiratory PCR panels. The authors should discuss this with more references cited, with these (PMID: 39857007 and 40137747) as example (citing is optional).

Please submit your revised manuscript by Jul 06 2025 11:59PM. If you will need more time than this to complete your revisions, please reply to this message or contact the journal office at plosone@plos.org . A rebuttal letter that responds to each point raised by the academic editor and reviewer(s). You should upload this letter as a separate file labeled 'Response to Reviewers'.A marked-up copy of your manuscript that highlights changes made to the original version. You should upload this as a separate file labeled 'Revised Manuscript with Track Changes'.An unmarked version of your revised paper without tracked changes. You should upload this as a separate file labeled 'Manuscript'.

We look forward to receiving your revised manuscript.

Kind regards,

Benjamin M. Liu, MBBS, PhD, D(ABMM), MB(ASCP)

Academic Editor

PLOS ONE

Journal Requirements:

3. We note that your Data Availability Statement is currently as follows: All relevant data are within the manuscript and its Supporting Information files

5. Please remove all personal information, ensure that the data shared are in accordance with participant consent, and re-upload a fully anonymized data set.

Reviewers' comments:

Reviewer's Responses to Questions

**Comments to the Author**

1. Is the manuscript technically sound, and do the data support the conclusions?

Reviewer #1: Yes

Reviewer #2: Partly

Reviewer #3: Yes

2. Has the statistical analysis been performed appropriately and rigorously?

Reviewer #1: Yes

Reviewer #2: Yes

Reviewer #3: Yes

3. Have the authors made all data underlying the findings in their manuscript fully available?

Reviewer #1: Yes

Reviewer #2: Yes

Reviewer #3: Yes

4. Is the manuscript presented in an intelligible fashion and written in standard English?

Reviewer #1: Yes

Reviewer #2: Yes

Reviewer #3: Yes

Reviewer #1: Thank you for sharing the manuscript. Below is a detailed peer review of the article “Body Mass Index Impact on ICU Interventions and Outcomes in Hospitalized Patients with COVID-19 Infection – A National Population-Based Study.”

SUMMARY OF REQUIRED CHANGES

Grammar: Revise awkward or passive constructions, eliminate redundancies

Ethics: Include IRB name, approval number or formal statement of exemption

References: Fix the citation error (!!! INVALID CITATION !!!), and ensure uniform formatting

Scientific Clarity: Explicitly address possible biases and explain clinical relevance of modest odds ratios

Consistency: Ensure consistent use of terms like “obesity,” “COVID-19 infection,” and “mechanical ventilation”

Reviewer #2: The manuscript addresses an important topic in hospitalized COVID-19 patients during the pre-vaccination period. The use of a large, national cohort from the HCA Healthcare database is a strength, providing robust statistical power. However, several methodological and interpretive concerns warrant attention before publication.

1. Innovation is not seen in the manuscript. As mentioned in lines 145-146, the effect on the complications of COVID-19 has been confirmed.

2. No statistics, tables, or comparisons have been provided or explained regarding various factors other than BMI. please adde to introduction and discussion.

3. Has body mass been examined alongside other factors such as the use of certain medications, underlying diseases, etc.? Please add it to the manuscript

4. Figure1 is very simple and the entire steps of sample selection, data processing, extraction, etc. need to be presented in the form of a tree.

5. The exclusion of patients with significant comorbidities (e.g., malignancy, chronic pulmonary disease, immunocompromised states) and DNR/DNI status is problematic. These exclusions may introduce selection bias, as these conditions are prevalent in real-world COVID-19 populations and could confound the relationship between BMI and outcomes. A sensitivity analysis including these patients, or a justification for their exclusion based on prior literature, is essential.

6. The R² values for multivariate linear regression models (0.053 for duration of mechanical ventilation and 0.037 for ICU LOS) are extremely low, indicating that the models explain only a small fraction of the variance. This suggests that critical unmeasured confounders (e.g., severity scores like SOFA or APACHE II, timing of interventions) may be missing. The reliance on BMI categories alone without adjusting for BMI as a continuous variable limits the granularity of findings.

7. The study relies on BMI calculated from medical records, which may be inaccurate due to supine height measurements, fluid status (e.g., in dialysis patients), or excess muscle mass. This introduces measurement error that could skew results, particularly for underweight and obese categories. The lack of discussion on these limitations undermines the reliability of conclusions.

8. The claim that obesity is independently associated with higher mortality (OR 1.29) is overstated given the non-significant difference in all-cause mortality across BMI categories (p=0.14, Table 3). This inconsistency suggests overfitting or residual confounding. The Discussion also overgeneralizes findings to the U.S. population without acknowledging regional healthcare disparities.

9. While the association between obesity and worse COVID-19 outcomes is consistent with prior studies (e.g., Kompaniyets et al., 2021), the manuscript fails to highlight novel insights or clinical interventions beyond existing knowledge. The recommendation for "tailored medical strategies" is vague and lacks specific actionable guidance.

10. The abstract is dense and lacks a concise summary of key findings (e.g., specific ORs or LOS differences). Revise to include a succinct results overview.

Reviewer #3: Review Comments

1. Structure and Title

The title is clear and effectively reflects the core content of the study. However, it could be slightly shortened for improved readability, for example:

"Body Mass Index and Critical Care Outcomes in Hospitalized COVID-19 Patients: A National Cohort Study"

Abstract: The abstract well-articulates the objectives, methodology, and key findings. However, in the Methods section, it would enhance clarity if the source of the data (e.g., specific database or registry) were explicitly mentioned.

2. Introduction

The introduction adequately sets the epidemiological and clinical context of the relationship between Body Mass Index (BMI) and respiratory disease severity, particularly in COVID-19 . However, more recent literature on the "obesity paradox" in respiratory illnesses could be referenced for a more comprehensive background

The hypothesis is clearly defined. Nevertheless, the significance of this study compared to previous work—such as the studies by Kompaniyets et al. (2021) and Kapoor et al. (2022) —should be more explicitly highlighted.

3. Methods

The study population and sample size (22,000 patients) are commendable, and data were drawn from the HCA Healthcare database, which is representative of 149 hospitals across 18 U.S. states. However, it would improve the methodological rigor if the authors clarified how variables such as prior vaccination status or previous SARS-CoV-2 infection (e.g., IgG serostatus) were accounted for in the analysis.

The BMI categories follow CDC standards, which is appropriate. However, the exclusion of the underweight group (BMI <18.5) from many analyses should be justified and discussed, especially if it was due to low sample size or confounding factors.

Statistical analysis: The use of ANOVA and regression models is suitable. However, further explanation should be provided regarding how confounding variables such as age, sex, and comorbidities (e.g., diabetes) were controlled in the models.

Limitations: The authors appropriately acknowledge limitations such as lack of access to detailed clinical data (e.g., inflammatory markers, medication use). However, this could be expanded by addressing how antiviral treatments and vaccination timing may have influenced outcomes, particularly during the pre-vaccination period.

4. Results

Table 3 is well-structured, but for the "Duration of Mechanical Ventilation" section, a bar chart would better emphasize differences than a tabular format.

Key Findings:

Obese patients had the highest rates of tracheostomy placement and prolonged mechanical ventilation, findings consistent with prior studies.

All-cause mortality did not show significant differences across BMI categories, which contrasts with some reported findings and deserves further discussion.

The multivariate analysis reported significant odds ratios for diabetes (OR: 1.70) , obesity (OR: 1.57), and chronic kidney disease (OR: 1.87) . However, the inverse association between age and tracheostomy need (OR: 0.98) requires clarification.

5. Discussion

The discussion compares well with previous studies such as Petrilli et al. (2020)

, but could benefit from referencing Simonnet et al. (2020), who demonstrated a strong association between obesity and mechanical ventilation requirements.

The clinical mechanisms linking obesity to mechanical ventilation needs are explained clearly. However, the role of altered leukocyte metabolism in obese individuals could be expanded upon to provide a deeper biological insight.

Limitations: The authors have appropriately addressed issues such as limited access to palliative care data and type of antiviral drugs used. Further elaboration on how obesity and associated comorbidities (e.g., diabetes) affected clinical decision-making and ICU management would strengthen this section.

Clinical Relevance: The results are well-connected to clinical practice, but the importance of weight management and obesity prevention strategies in reducing risks associated with COVID-19 severity should be emphasized.

6. Conclusion

The conclusions are well-summarized. To strengthen the impact, the authors could specify actionable strategies for managing obese patients with COVID-19, such as:

Increased vigilance in respiratory support.

Early decision-making protocols for tracheostomy when indicated.

Prioritizing vaccination and early intervention in obese patient groups.

7. Tables and Figures

Table 3 is well-constructed, but the variables "Need for Mechanical Ventilation" and "Tracheostomy Placement" would benefit from being presented as percentages to improve interpretability.

Figures 3 and 4 are appropriately structured, but the legends should clearly indicate the confidence intervals (CI) and beta/OR values for the regression outputs.

Figure Captions: The captions could be more precise. For instance, in Figure 4, it would improve clarity to specify the BMI range for the obese category.

8. References

The reference list includes recent and relevant sources. However, additional citations could be added for topics such as obesity and cytokine storm, and immunometabolism in obesity.

Some references, particularly

, should be updated to include the correct author names and publication years.

**Do you want your identity to be public for this peer review?** For information about this choice, including consent withdrawal, please see our Privacy Policy

Reviewer #1: **Yes: ** Ali Khanifar

Reviewer #2: **Yes: ** Davood Azadi

Reviewer #3: No

---

## [Author Response · Author response to Decision Letter 1]

18 Jul 2025

Reviewer #1

Comment 3 (C3): Missing Ethics Details. The ethics section says "N/A" yet claims IRB exemption later in the methods. Recommendation: Clearly name the IRB, approval number, and exemption reason as per PLOS ONE guidelines.

Response 3 (R3): We apologize for this oversight. We have expanded on the ethics section to be concordant with our ethical considerations section of the manuscript. Additionally, we have added further details of the IRB, approval number and exemption reason as per PLOS ONE guidelines.

Comment 4 (C4): Selection bias. Exclusion of DNR/DNI patients and patients with certain comorbidities may introduce survivorship bias. Recommendation: Discuss this as a limitation.

Response 4 (R4): We appreciate this input. We have expanded on this limitation in the discussion section of the manuscript.

Comment 5 (C5): BMI measurement error. BMI in critical care can be innacurate due to fluid shifts or supine posititioning. Recommendation: Address in limitations more explicitely.

Response 5 (R5): We thank the reviewer, and have addressed this limitation more explicitely in the discussion section of our manuscript.

Comment 6 (C6): Temporal bias. The study uses 2020 data. The evolving nature of treatment protocols during that year may influence results. Recommendation: Consider stratifying analysis by quarter/month if possible

Response 6 (R6): We appreciate the insight provided by the reviewers and are in agreement. We have added this as a limitation to our study in the discussion section. Although our selection criteria aim to capture a pre-vaccine population, we did not control for the ongoing treatment protocols during March to December 2020 due to its variability and inconsistency in practice across hospitals. Unfortunately, we are unable to stratify by quarters our data analysis at this point.

Comment 7(C7): Effect Sizes. Although ORs are significant, many are modest (e.g., OR ~1.3). Recommendation: Discuss clinical versus. statistical significance.

Response 7 (R7): We appreciate the reviewer’s thoughtful comment. We recognize that some of the observed odds ratios (e.g., OR ~1.3) are modest in magnitude. While these values are statistically significant due to our large sample size, we acknowledge that they may have limited individual-level clinical impact. However, we believe that the consistency of these associations across diverse geographic regions, combined with the population-level implications, underscores their clinical relevance. Even modest increases in risk, when applied to large populations—particularly during a global pandemic—can result in substantial added strain on healthcare systems and contribute meaningfully to public health planning. We have added a paragraph in our discussion to highlight the authors’ opinion.

Comment 8 (C8): Verb Agreement & Sentence Structure: Example: “Our study revealed a higher utilization and duration of respiratory support in the ICU.” → Consider splitting or rephrasing. Correction: “Our study revealed both higher utilization and longer duration of respiratory support in the ICU”.

Response 8 (R8): We appreciate the attention our reviewers have given to our manuscript’s grammar and sentence construction. We had proceeded to incorporate the correction. Additionally, we have reviewed in depth the grammar and sentence construction of the entire manuscript and edited it accordingly to enhance the delivery of a clear and concise message to the reader.

Comment 9 (C9): Repetitive Phrasing: Many parts redundantly explain the same physiological mechanisms (e.g., ACE-2 expression, cytokine storm). Suggest condensing to improve readability.

Response 9 (R9): We appreciate the input. We have condensed this section to enhance readability and grouped it with related sentences in a specific paragraph within the manuscript’s discussion.

Comment 10 (C10): Overuse of “Obese patients were found…”: Consider varying language: e.g., "Individuals in the obese category showed...", "Participants with obesity demonstrated..."

Response 10 (R10): We are thankfull to the reviewer for noticing this overuse of words. We have revised the full manuscript to ensure that it is no longer the case.

Comment 11 (C11): Inconsistent Use of Terms: Sometimes uses “COVID-19 infection” and other times just “COVID-19.” Recommend consistent usage.

Response 11 (R11): We appreciate the detailed review of our manuscript. We have identified this inconsistency and corrected it throughout the entire manuscript.

Comment 12 (C12): Ambiguous Wording: Example: “Patients classified as obese or overweight were noted to have extended stays.” → Use active voice: “Obese and overweight patients had longer ICU stays.”

Response 12 (R12): We value the reviewer’s input. We have removed the ambiguous wording from our discussion section and believe we have eased the flow of the discussion and enhanced clarity.

Comment 13 (C13): Citation Error: There is an “!!! INVALID CITATION !!!” placeholder under references. Must be corrected before resubmission.

Response 13 (R13): We apologize for this oversight. We had removed the wrongly entered references from our manuscript.

Reviewer #2

Comment 14 (C14): Innovation is not seen in the manuscript. As mentioned in lines 145-146, the effect of the complications of COVID-19 has been confirmed.

Response 14 (R14): We acknowledge the insight from our reviewer and understand the reviewer’s stand. We, the authors, believe that our study has contributed to the medical community by confirming previously investigated outcomes in a larger and geographically diverse patient population within a broad hospital system, thereby providing additional scientific value.

Comment 15 (C15): No statistics, tables, or comparisons have been provided or explained regarding various factors other than BMI. Please add to the introduction and discussion.

Response 15 (R15): We thank the reviewer for the observation. We highlight gender, race, and ethnicity, finding differences within the discussion section of our manuscript while focusing primary of BMI categories as the variable of interest.

Comment 16 (C16): Has body mass been examined alongside other factors such as the use of certain medications, underlying diseases, etc.? Please add it to the manuscript.

Response 16 (R16): We appreciate the input. As also pointed out by another reviewer, our study did not adjust for other co-founders, specifically emergent interventions or specific pathologies, during the pre-vaccination pandemic era. We have added it to our discussion as a study limitation, specifically addressing temporal bias and survivorship bias, in response to both reviewers’ suggestions.

Comment 17 (C17): Figure 1 is very simple, and the entire steps of sample selection, data processing, extraction, etc., need to be presented in the form of a tree.

Response 17 (R17): We appreciate the feedback. We have expanded the sample selection steps in Figure 1 and presented them in a more user-friendly format.

Comment 18 (C18): The exclusion of patients with significant comorbidities (e.g., malignancy, chronic pulmonary disease, immunocompromised states) and DNR/DNI status is problematic. These exclusions may introduce selection bias, as these conditions are prevalent in real-world COVID-19 populations and could confound the relationship between BMI and outcomes. A sensitivity analysis including these patients, or a justification for their exclusion based on prior literature, is essential.

Response 18 (R18): Thank you to the reviewer for highlighting this important point brought up to the authors. We agree with the reviewer's observation, which was also mentioned by other reviewers. The survivorship bias concern has been placed as a limitation of the study. Additionally, we have a statement in the methodology section, under patient population, to share the authors’ decision-making process for excluding patients with significant comorbidities.

Comment 19 (C19): The R² values for multivariate linear regression models (0.053 for duration of mechanical ventilation and 0.037 for ICU LOS) are extremely low, indicating that the models explain only a small fraction of the variance. This suggests that critical unmeasured confounders (e.g., severity scores like SOFA or APACHE II, timing of interventions) may be missing. The reliance on BMI categories alone, without adjusting for BMI as a continuous variable, limits the granularity of the findings.

Response 19 (R19): We appreciate the reviewer’s observation and agree on the shortcomings of the models. Potential unmeasured confounders, such as those mentioned by our reviewers, could weaken our findings; however, our dataset did not allow for these to be accurately calculated. We have included this as an important limitation of our study.

Comment 20 (C20): The study relies on BMI calculated from medical records, which may be inaccurate due to supine height measurements, fluid status (e.g., in dialysis patients), or excess muscle mass. This introduces measurement error that could skew results, particularly for underweight and obese categories. The lack of discussion on these limitations undermines the reliability of conclusions.

Response 20 (R20): We agree with the reviewer. As also suggested by a different reviewer, this inaccuracy is challenging to address due to the nature of our study's methodology. We have proceeded to explain and expand on this inaccuracy as a limitation of our research in the discussion section.

Comment 21 (C21): The claim that obesity is independently associated with higher mortality (OR 1.29) is overstated given the non-significant difference in all-cause mortality across BMI categories (p=0.14, Table 3). This inconsistency suggests overfitting or residual confounding. The Discussion also overgeneralizes findings to the U.S. population without acknowledging regional healthcare disparities.

Response 21 (R21): We thank our reviewer for the insight. We have reformatted the discussion on the association between obesity and mortality, de-emphasizing a strong association and commenting on the lack of difference in the frequency of all mortality or hospice admissions cases among BMI groups. We also address the low odds ratio (OR) to highlight clinical over statistical significance. We have included our analysis of inpatient (hospice excluded) mortality for completion. Additionally, we comment on regional health care disparities. The authors believe that the dataset allows for a certain degree of generalization due to the large number of datapoints and the inclusion of several healthcare system regions, which are distinctive from one another, that contributed to the dataset.

Comment 22 (C22): While the association between obesity and worse COVID-19 outcomes is consistent with prior studies (e.g., Kompaniyets et al., 2021), the manuscript fails to highlight novel insights or clinical interventions beyond existing knowledge. The recommendation for "tailored medical strategies" is vague and lacks specific, actionable guidance.

Response 22 (R22): We thank the reviewer and remove this recommendation, understanding the lack of specific and actionable guidance.

Comment 23 (C23): The abstract is dense and lacks a concise summary of key findings (e.g., specific ORs or LOS differences). Revise to include a succinct results overview

Response 23 (R23): We recognize the merit of the reviewer’s critique. We have revised the abstract in detail and in its entirety and provided a succinct yet comprehensive description of our study design and findings.

Reviewer #3

Comment 24 (C24): Structure and Title The title is clear and effectively reflects the core content of the study. However, it could be slightly shortened for improved readability, for example: "Body Mass Index and Critical Care Outcomes in Hospitalized COVID-19 Patients: A National Cohort Study"

Response 24 (R24): We value the insight regarding our manuscript title and have adjusted the manuscript to reflect the reviewer’s feedback.

Comment 25 (C25): Abstract: The abstract well-articulates the objectives, methodology, and key findings. However, in the Methods section, it would enhance clarity if the source of the data (e.g., specific database or registry) were explicitly mentioned.

Response 25 (R25): We thank you the reviewer for the recommendation. We had enhance the clarity of the methods section of the abstract by restructuring the description of the database used for our study.

Comment 26 (C26): The introduction adequately sets the epidemiological and clinical context of the relationship between Body Mass Index (BMI) and respiratory disease severity, particularly in COVID-19 . However, more recent literature on the "obesity paradox" in respiratory illnesses could be referenced for a more comprehensive background

Response 26 (R26): We appreciate the valuable information and the referent to the obesity paradox in respiratory illnesses. We have included this key information and related to our study in the background section of the manuscript.

Comment 27 (C27): The hypothesis is clearly defined. Nevertheless, the significance of this study compared to previous work—such as the studies by Kompaniyets et al. (2021) and Kapoor et al. (2022) —should be more explicitly highlighted.

Response 27 (R27): We appreciate the input and proceeded to highlight both studies more prominently in our background section leading to our study aim.

Comment 28 (C28): The study population and sample size (22,000 patients) are commendable, and data were drawn from the HCA Healthcare database, which is representative of 149 hospitals across 18 U.S. states. However, it would improve the methodological rigor if the authors clarified how variables such as prior vaccination status or previous SARS-CoV-2 infection (e.g., IgG serostatus) were accounted for in the analysis.

Response 28 (R28): We appreciate the feedback provided by the reviewer and include a statement in our methods section to improve methodology rigor in the variable of interest sub-section.

Comment 29 (C29): The BMI categories follow CDC standards, which is appropriate. However, the exclusion of the underweight group (BMI <18.5) from many analyses should be justified and discussed, especially if it was due to low sample size or confounding factors.

Response 29 (R29): We value the input provided. We opted to focus on the comparison of the obese patient with the normal BMI individual based on the supporting evidence of adverse related outcomes. Although the low BMI patient group carried a n>500, our predictive models did not fnd any association with the outcomes of interest. In fight of keeping the manuscript targets and under the word limit requirements, we opted to exclude this from the discussion. If the reviewer thinks it is fundamental to add a statement in the discussion on the low BMI group findigns described on the tables and figures, we could definitely consider it and the authors are willing to implement it upon request.

Comment 30 (C30): Statistical analysis: The use of ANOVA and regression models is suitable. However, further explanation should be provided regarding how confounding variables such as age, sex, and comorbidities (e.g., diabetes) were controlled in the models.

Response 30 (R30): We thank the reviewer for the opportunity to enhance our manuscript. In our multivariable regression models, we adjusted for key potential confounders, including age, sex, and relevant comorbidities such as diabetes, hypertension, coronary artery disease, and chronic kidney disease. These variables were selected based on established clinical relevance and prior literature indicating their impact on COVID-19 outcomes. By including them as covariates in the models, we aimed to isolate the independent association between body mass index (BMI) and clinical outcomes. We have revised the Methods section of the manuscript to clearly describe this adjustment approach.

Comment 31 (C31): Limitations: The authors appropriately acknowledge limitations such as lack of access to detailed clinical data (e.g., inflammatory markers, medication use). However, this could be expanded by addressing how

---

## [Editor Report · Decision Letter 1]

22 Jul 2025

Body mass index and critical care outcomes in hospitalized COVID-19 patients – a national cohort study

PONE-D-25-18274R1

Dear Dr. Bonilla,

We’re pleased to inform you that your manuscript has been judged scientifically suitable for publication and will be formally accepted for publication once it meets all outstanding technical requirements.

Kind regards,

Benjamin M. Liu, MBBS, PhD, D(ABMM), MB(ASCP)

Academic Editor

PLOS ONE
---

## [Editor Report · Acceptance letter]

PONE-D-25-18274R1

PLOS ONE

Dear Dr. Bonilla,

I'm pleased to inform you that your manuscript has been deemed suitable for publication in PLOS ONE. Congratulations! Your manuscript is now being handed over to our production team.

Kind regards,

on behalf of

Dr. Benjamin M. Liu

Academic Editor

PLOS ONE